# Anti-Biofouling Strategies for Long-Term Continuous Use of Implantable Biosensors

**Jian Xu**  **and Hyowon Lee \***

Weldon School of Biomedical Engineering, Birck Nanotechnology Center, Center for Implantable Devices, Purdue University, West Lafayette, IN 47907, USA
* Correspondence: hwlee@purdue.edu

**Abstract:** The growing trend for personalized medicine calls for more reliable implantable biosensors that are capable of continuously monitoring target analytes for extended periods (i.e., >30 d). While promising biosensors for various applications are constantly being developed in the laboratories across the world, many struggle to maintain reliable functionality in complex in vivo environments over time. In this review, we explore the impact of various biotic and abiotic failure modes on the reliability of implantable biosensors. We discuss various design considerations for the development of chronically reliable implantable biosensors with a specific focus on strategies to combat biofouling, which is a fundamental challenge for many implantable devices. Briefly, we introduce the process of the foreign body response and compare the in vitro and the in vivo performances of state-of-the-art implantable biosensors. We then discuss the latest development in material science to minimize and delay biofouling including the usage of various hydrophilic, biomimetic, drug-eluting, zwitterionic, and other smart polymer materials. We also explore a number of active anti-biofouling approaches including stimuli-responsive materials and mechanical actuation. Finally, we conclude this topical review with a discussion on future research opportunities towards more reliable implantable biosensors.

**Keywords:** biosensor; implantable; biofouling; foreign body response; non-specific binding

---

## 1. Introduction

Since their invention more than 50 years ago [1], biosensors have been one of the most actively researched areas in biomedical science. Commercially, the market for biosensors has sustained a 7.5% average annual growth rate for decades [2], and was valued at USD 19.6 billion in 2019. This trend is further highlighted by the recent proliferation of personalized medicine, point-of-care testing, and wearable devices [3]. The wide-ranging utility of biosensors in clinical care and biomedical research is well recognized. Examples of biosensor uses include helping diabetic patients monitor their blood glucose levels, warning surgeons about an impending circulatory failure [4], and enabling investigation on the role of neurotransmitters in neurodegeneration and affective disorders [5–9].

Biosensors are analytical devices designed for the detection of analytes, combining a biological component with a physicochemical detector [10,11]. Based on this definition, every biosensor has two major components:

- Biorecognition component
- Transducing component

The biorecognition component provides a high selectivity between the specific analyte and the recognition unit. This process usually involves either a catalytic reaction (enzyme-based) or the capturing of molecules (bioaffinity-based) using antibodies, antigens, aptamers, or deoxyribonucleic/

ribonucleic acid (DNA/RNA) segments. Based on the presence and concentration of the target analyte, a measurable signal such as change in mass, film thickness, charge, dielectric constant, or fluorescence can be generated [12]. Depending on the type of signal transduction mechanism, a biosensor can be categorized into types as shown in Table 1 [13–15].

**Table 1.** Types of biosensor transducers and corresponding signals during operation.

| Transducer Types | | Measured Signals | Example |
|---|---|---|---|
| Electrochemical | Amperometric | Changes in current when applying a fixed potential between reference and working electrode. | [16] |
| | Conductometric | Changes in electrical conductivity of the medium between two electrodes. | [17] |
| | Impedimetric | Changes in impedance (magnitude and/or phase) over a wide range of alternating current (AC) frequencies. | [18] |
| | Potentiometric | Changes in zero-current potential between a reference electrode and working electrode. | [14,19] |
| | Voltammetric | Changes in resulting current when varying potential applied to the working electrode. | [15] |
| Optical | Surface plasmon resonance | Changes in light absorption, reflectance, fluorescence, Raman scattering (RS), or refractive index (RI). | [20,21] |
| | Optical waveguides | | |
| | Optical resonators | | |
| | Photonic crystals | | |
| | Optical fibers | | |
| Field-effect transistor (FET) based | | Changes in current between the source and drain electrodes due to the electrostatic surface potential change of the semiconductor. | [22,23] |
| Organic electrochemical transistor (OECT) based | | Change in current due to the ion injections from the electrolyte of interest into a semiconductor channel. | [24] |
| Piezoelectric | | Changes in resonance frequency due to mass change of a piezoelectric crystal. | [25] |
| Thermometric | | Changes in temperature induced by the biological reactions. | [26,27] |
| Magnetic | | Changes in magnetic field or magnetically induced effects. | [28,29] |

In addition to the biological recognition and transducing components, the operation of a biosensor often requires auxiliary parts including power source, potentiostat, signal processing unit, data storage, and telemetry system [30]. Biosensors are often characterized using the following metrics:

1.　Sensitivity
2.　Limit of detection
3.　Selectivity
4.　Accuracy

High sensitivity ensures that the biosensors have the ability to detect the target analyte in clinically-relevant or diluted concentrations [12]. Commonly, the sensitivity $S$ is defined as the ratio between the output signal and the concentration of the analyte:

$$S = \frac{\partial f([a])}{\partial [a]} \approx \frac{f([a]) - f(0)}{[a]}, \tag{1}$$

where $f$ is the function of sensor output with respect to analyte concentration $[a]$. This approximation is valid for biosensors with linear response.

The limit of detection (LOD) is the minimal change in concentration that the sensor can confidently distinguish from the absence of the analyte:

$$\text{LOD} = \frac{3\sigma}{S}, \tag{2}$$

where $\sigma$ is the standard deviation of the baseline noise when $[a] = 0$. In comparison, the limit of quantification (LOQ) is also used sometimes to describe the lowest analyte concentration that can be quantitatively detected with a predefined accuracy and precision. The determination of LOQ is set by biosensor developers, and its value is always higher than LOD [31]. The dynamic range of the biosensor refers to the operational window between the upper and lower limits of measurable analyte concentration.

The selectivity refers to the biosensors' capability to transduce the specific target analyte in the presence of other biological components. A high selectivity may be achieved using highly specific recognition elements, permselective membranes (i.e., Nafion), and by incorporation of nanomaterials (i.e., carbon nanotubes) [32,33]. The selectivity of the biosensor can be measured by introducing analogous analytes typically found in the working environment of the biosensor and comparing the magnitude of the sensor response versus the actual target. Minimal response to analogous analyte indicates a better selectivity.

Since the outputs of biosensors are often directly related to clinical diagnosis and therapy, their accuracy is critical. The accuracy of a sensor is typically quantified as the mean absolute relative difference (MARD) between the measured and the real values [34]:

$$\text{MARD} = \frac{1}{N} \sum_i^N \left| \frac{f_{sensor}(t_i) - f_{ref}(t_i)}{f_{ref}(t_i)} \right|. \tag{3}$$

MARD represent the average variation in measurement between the sensor ($f_{sensor}(t_i)$) and the true reference value ($f_{ref}(t_i)$) across the sample population ($N$), with lower values suggesting a higher accuracy. A high MARD value of a biosensor may be an indication of low selectivity, high noise, persistent bias, or other confounding factors [12].

In addition to these four fundamental metrics, the performance of a biosensor can also be characterized by its response time, durability, reversibility, lifetime, and spatiotemporal resolution depending on the sensor application. For example, it is essential for chronically implantable biosensors to have minimum signal drift and high long-term reliability to improve its utility. The issue of signal degradation and sensor reliability over time is often attributed to biofouling.

Biofouling is defined as spontaneous accumulation of macromolecules or microorganisms (e.g., proteins, cells, bacteria). The adsorption of these biofouling materials can physically limit the diffusion of the target analyte to the sensor surface. Moreover, the accumulated proteins could further trigger the foreign body response (FBR), leading to gradual encapsulation of the sensor, blocking analyte access, and ultimately causing sensor failure [35,36].

In this review, we explore various emerging strategies to prevent or mitigate biofouling for implantable biosensor applications. First, we briefly discuss the unique design considerations associated with the development of implantable biosensors. We then discuss various anti-biofouling strategies including passive approaches (e.g., use of hydrophilic surfaces, zwitterionic polymers, naturally-occurring or biomimicking materials, superhydrophobic surfaces, and drug-eluting materials), and active approaches (e.g., use of temperature or pH-responsive materials, acoustic waves, mechanical actuation, and surfactant-desorbing surfaces). Improving the functional lifetime of these implantable sensors using these state-of-the-art anti-biofouling strategies may result in the next generation of biosensors for novel clinical diagnostic and therapeutic opportunities.

## 2. Implantable Biosensors and Their Design

Working within the body, implantable biosensors have the ability to continuously monitor patients without intervention for an extended period of time [37]. While the management of blood glucose levels for diabetic patients remains a top driving force for the development of implantable biosensors, the monitoring of other targets such as lactate, neurotransmitters, chemotherapeutics and anti-cancer medication, and heavy metal ions has also been realized [12,38–40]. The key advantages of implantable biosensors include:

- Patient comfort: the monitoring of analyte via implantable biosensors does not typically require further clinical intervention once implanted and is continuous regardless of patient activity (e.g., rest, sleep, exercise) [32].
- Extended information: implantable biosensors often provide a higher temporal resolution with continuous monitoring that can reveal hidden patterns, which could be vital for early clinical diagnosis and therapy [41–43].
- Expandability: implantable biosensors may be integrated with a drug delivery mechanism to enable a closed-loop system for disease management. For example, type I diabetic patients can receive insulin in automatically adjusted doses [44], cancer patients can ensure their chemotherapeutics are maintained within narrow therapeutic windows [45], chronic pain patients can receive pain medication bolus as needed [46].
- New research tool: through real-time detection of neurotransmitters or electric signals, implantable biosensors could offer tools for fundamental neuroscience and behavioral research [5,7].

Compared to other permanent or semi-permanent implantable devices such as catheters, pacemakers, heart valves, or stents, existing implantable biosensors have a considerably shorter lifetime (e.g., weeks vs. years) [47]. The limited lifetime is often due to the increased risk of failure in the complex in vivo environment. For all implantable biosensors, their failure mechanisms may be put into two categories [35,36]:

- Abiotic failure such as electrode corrosion and detachment, insulation delamination, and electrical short.
- Biotic failure such as membrane biofouling, biorecognition element inactivation, passivation, and fibrous encapsulation.

Although these are seemingly two distinguishable failure modes, the initiation of most abiotic failure is often due to the in vivo environment (pH, temperature, various biomolecules, ions, electrolytes, etc.). Some early studies even found that biosensors failed in vivo may regain their functionality once they were explanted and tested again in vitro [48,49]. Hence, implantable biosensors require unique design considerations in order to maintain their performance (e.g., sensitivity, selectivity, accuracy) over an extended period of time in vivo.

### 2.1. Design Consideration

#### 2.1.1. Biorecognition Component

A common biotic failure mode of implantable biosensors is related to the degradation, inactivity, depletion, and the desorption of their biorecognition units during the operation. Therefore, a stable biorecognition component is essential for long-term accurate monitoring of the interested analytes. The overwhelming majority of implantable biosensors use enzymes as their biorecognition component [16]. When used at their optimal temperature and pH, enzymes are quite stable. However, they can degrade metabolically.

Conversely, biological recognition units in bioaffinity-based biosensors are usually consumable due to the irreversible binding to the analytes. These recognition units are gradually used up once

available receptor sites get depleted, severely limiting the lifetime if used as an implantable biosensor. However, depending on binding kinetics, some bioaffinity-based biosensors can be regenerated and used for a longer period of time via electrochemical activation or the incorporation of an electrolyte flow system [50,51]. For these reasons, bioaffinity-based biosensors are more often used in point-of-care diagnostics.

One issue of developing enzymatic biosensors for in vivo application is their dependence on oxygen ($O_2$) and hydrogen peroxide ($H_2O_2$). First-generation oxidase or dehydrogenase enzymes require the presences of dissolved $O_2$ for their specific reactions with analytes, and the signal generation of these enzymatic biosensors requires the oxidation of $H_2O_2$. Several studies have demonstrated that implanted biosensors often provide inaccurate data when the subject is under anesthesia, which lowers systemic oxygen levels [52,53]. Moreover, the inflammatory response due to the implant could also lower local $O_2$ and $H_2O_2$ levels that can affect sensor performance [54–56]. The dependence of $O_2/H_2O_2$ may be avoided by using enzymes paired with mediators (second-generation), such as ferrocene derivatives, ferricyanide, conducting organic salts, quinone compounds, transition metal complexes [57], or using direct electron transfer (third-generation) [58,59]. However, the stability and possible toxicity of mediators need to be considered carefully.

Another special consideration for implantable enzymatic biosensors is the enzyme immobilization. As proteins, enzymes can denature metabolically and lose functionality in the host body. The loss of enzyme activity over time leads to decreased sensitivity and selectivity, making it one of the most common biotic failure modes for implantable biosensors [60,61]. Generally, dried enzymes are more stable than its solubilized form, and immobilized enzymes often have better activity and stability [61].

Having a stable enzyme layer with minimum desorption and degradation can prolong the biosensor lifetime. Over the years, various enzyme immobilization methods have been developed including the use of non-specific adsorption, sol-gel process, covalent binding, and polymeric films [35]. A more detailed discussion on the topic of enzyme immobilization can be found in the following review papers [62–64].

### 2.1.2. Transducing Component

The signal transducing component of implantable biosensors also requires a robust design to prevent premature failure. Studies have suggested the physical and mechanical properties of implants play a role in mitigating or exacerbating the host response to them. Hard, sharp, and rough implant surfaces result in a more severe inflammatory response, greater activation of macrophages, and thicker fibrotic capsulation, which all contribute to the decreased sensitivity and selectivity of biosensors [65]. Moreover, the implantable biosensors are expected to be under low-magnitude but constant strain and motion [66]. This constant mechanical stress and fatigue may lead to mechanical failures such as broken wires, cracked casings, or shorted electrodes. Hence, the implantable sensors need to be physically robust, flexible, or mechanically compatible with the native tissue.

Due to their simplicity, the majority of implantable biosensors are electrochemical [12,13]. The earliest reported implantable biosensors are enzyme-based amperometric biosensors (Table 1) [67,68]. Typical electrochemical biosensors consist of three electrodes: working, counter, and reference electrodes. Working electrodes are functionalized specifically to the analyte of interest via immobilization of the biorecognition component. By taking advantage of the development of micro-electromechanical systems (MEMS), electrochemical biosensors can now easily be miniaturized. Lately, driven by the demand for continuous glucose monitoring (CGM) and closed-loop insulin therapy, various implantable glucose biosensors have been developed and commercialized. The latest available CGMs in the US market such as Medtronic Guardian™ [69], Dexcom G6 [70], and Abbott FreeStyle Libre [71] are all enzyme-based amperometric implantable biosensors.

Alternatively, fluorescent biosensors also have been used in implantable applications. Such biosensors use fluorescence-conjugated materials, which react to the target analytes. Fluorescent signal is both excited and recorded at the site of the implant, and a wireless signal is transmitted and conveys the analyte concentration. The newly approved Eversense by Senseonics is an implantable fluorescent biosensor and capable of up to 90 d of continuous monitoring [72]. It consists of polymethyl methacrylate (PMMA) encasement, electronics, optical system, glucose-sensitive hydrogel, and a silicone collar eluting anti-inflammatory steroids (dexamethasone acetate) [73]. Its lifetime is considerably longer than amperometric competitors. However, it requires a clinical implantation due to the size of the sensor ($3.5 \times 18.3$ mm$^2$). Another limitation of this optical biosensor is the need for daily calibration using a gold-standard finger-prick method [12,74].

## 2.2. Biofouling and Foreign Body Response

Before being used clinically, the biocompatibility of implantable biosensors needs to be evaluated, which involves the evaluation of

1. Biosafety (i.e., the level of systemic and local host response to the implants including cytotoxicity, mutagenicity, or carcinogenicity);
2. Biofunctionality (i.e., the ability of implants to perform the designed task for certain period of time).

Upon incision and implantation of a device, a cascade of immune responses is triggered including the intrinsic and extrinsic coagulation cascades, the thrombus formation, the complement system, and the fibrinolytic system. Together, they make up the inflammatory or foreign body response (Figure 1) [16,75,76]. These inflammatory responses are initiated by the protein adsorption on the device surface within the first few seconds after the implantation, followed by the development of a blood-based transient provisional matrix. In the next few days, this protein-enriched matrix regulates the immune response by recruiting various types of immune cells to the surrounding tissue, including neutrophils, monocytes, and mast cells. In the following weeks, more immune cells infiltrate and release cytokines (i.e., interleukin-4 (IL-4)/IL-13) for further cell signaling. During this process, these cells either undergo activation and turn into macrophages or fuse into a foreign body giant cell (FBGC) to phagocytose foreign materials. In addition, collagens and fibroblasts are recruited to start tissue repair. The end result is either a complete integration of implant into the host, or the development of a dense, hypocellular, hypopermeable, hypovascular, and collagen-rich capsule to wall off the implant [40]. The latter case is extremely unfavorable for implantable biosensors that require direct access to the analyte-rich physiological fluid [77]. The evaluation of biosafety is relatively easy and can be done in vitro by examining the viability and proliferation of cells in culture. However, the assessment of the biofunctionality of implantable biosensors is a task as complicated as the biological composition of the host. For the most accurate evaluation of the functionality of implanted sensors, in vivo studies are necessary. During these studies, actual biosensors are implanted in animals (e.g., mice, rats, dogs, chimpanzees, and even humans) and examined for their long-term performance. These studies are expensive, time-consuming, labor-intensive, animal model dependent, and require dedicated physiological and histological support [78].

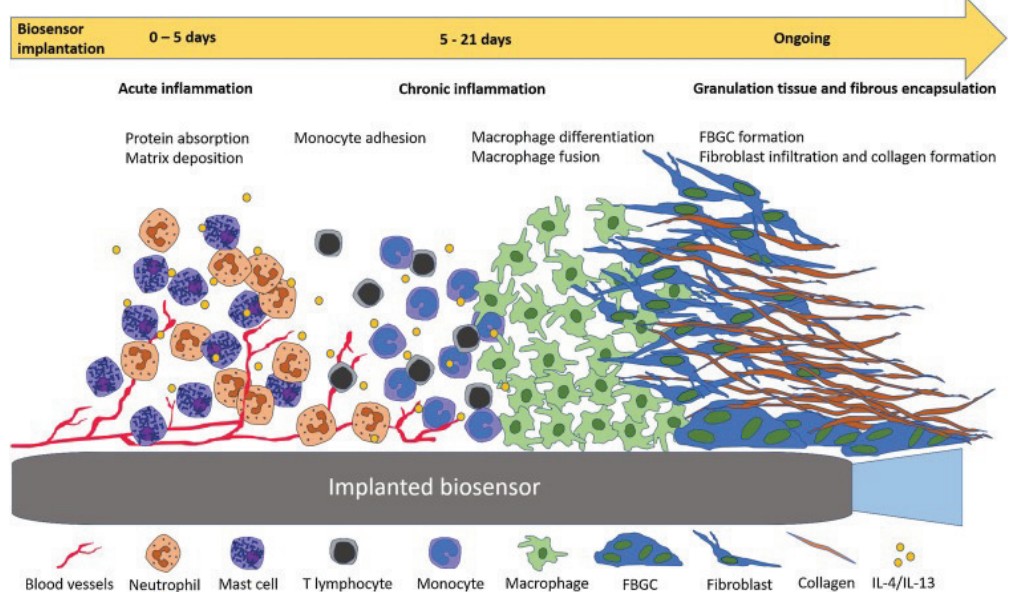

**Figure 1.** Diagram illustrating the inflammation and foreign body response to the implantable biosensors, which involves the adsorption of proteins, recruitment of immune cells, and development of fibrous tissue. Reprinted with permission from [40].

To minimize the number of animals required to evaluate the biofunctionality of implants during their development, it is also possible to use various analogous in vitro models to recapitulate parts of the immune systems. For example, researchers could gain valuable insights about the biocompatibility of their implantable biosensors by studying their ability to prevent and mitigate the attachment of proteins, bacteria, and cells. Non-specific binding of proteins to the implant surfaces occurs nearly instantaneously upon contact with blood and interstitial fluids. These initially bound proteins (e.g., fibrinogen and albumin) determine and regulate the cascade of immune responses [75,79]. Hence, the level of protein adsorption onto the implant surfaces is commonly tested. At the same time, the adhesion and the growth of bacteria on implant surfaces can also be examined to address potential infection and biofilm-induced chronic immune response [80].

In the remainder of this review work, we will present and discuss different anti-biofouling strategies that have been used for implantable biosensors. These strategies will be categorized into passive prevention of incoming biofouling or active removal of already-adsorbed fouling elements. Studies that expand out of in vitro testing will be highlighted, including inflammation modulation and long term in vivo studies. A summary of selected studies can be found in Table 2 for comparison.

**Table 2.** Selected studies of various anti-biofouling strategies for implantable biosensor application.

| Anti-Biofouling Strategies | | Biosensor Characterization | | | In Vitro Biofouling Tests | In Vivo Tests | Ref. |
|---|---|---|---|---|---|---|---|
| | | Analyte | Type | Calibration Results (Compared to Control) | | | |
| Passive approaches | PEG | Glucose | Enzymatic voltammetric | High sensitivity in whole blood, selectivity. | Suppressed blood fibrins and cell adsorption. | - | [81] |
| | Hydrogel | Glucose | Enzymatic amperometric | Improved sensitivity, response time, and linearity. | - | Functioned subcutaneously for at least 21 d. | [82] |
| | Zwitterionic polymer | Glucose | Enzymatic amperometric | Improved sensitivity and long-term stability. | Resistance to fibrinogen and human blood plasma adsorption. | - | [83] |
| | Naturally-occurring collagen | Glucose | Enzymatic amperometric | Similar sensitivity, and long-term stability. | - | Reduced inflammation, subcutaneously functional for 28 d. | [84] |
| | Naturally-occurring peptide | Immuno- globulin E (IgE) | Aptamer voltammetric | Improved sensitivity, LOD, and linear range, high selectivity. | Retained response in 0, 1, 2, and 5% fetal bovine serum. | - | [85] |
| | Superhydrophobic | Endothelial cells | Antibody fluorescent | Successfully detected targeted cells. | Resistance to proteins, platelets, human plasma and whole blood. | - | [86] |
| | Anti-inflammatory drug-eluting | Glucose | Enzymatic fluorescent | Similar sensitivity and linear range. | Retained response in simulated interstitial fluid, no cytotoxicity. | No sign of inflammation after 28 d of subcutaneous implantation. | [87] |
| | Angiogenic drug-eluting | Glucose | Enzymatic amperometric | - | - | Extended lifetime, faster and more accurate response subcutaneously. | [88] |
| Active approaches | Temperature-responsive | - | - | - | - | Thinner capsulation and increased microvascular after 30 d. | [89] |
| | Acoustic waves | Immuno- globulin G (IgG) | Antibody fluorescent | Detected specific bound proteins via frequency shifts. | Selectively removed non-specific bound proteins after 10 min actuation. | - | [90] |
| | Magnetic actuator | - | - | - | Removed 85% of bovine serum albumin (BSA) after 5 min actuation. | - | [91] |

## 3. Passive Anti-Biofouling Strategies

In this section, we will discuss various passive anti-biofouling strategies to combat against biofouling on implantable biosensors. These strategies include the use of naturally occurring materials, hydrophilic materials, superhydrophobic materials, and drug-eluting materials. Their working mechanisms vary from rendering surfaces thermodynamically unfavorable for fouling attachment, promoting the releasing of proteins, imitating native biological systems, and regulating host immune response.

### 3.1. Hydrophilic Materials

In aqueous environments, proteins undergo structural folding to better interact with water. Thermodynamically, the interaction between the hydrophilic outer shell of proteins and hydrophilic materials is more reversible because their interactions are entropically unfavorable [51]. Molecular simulation has also revealed that hydrophilic materials exhibit a stronger interaction with water molecules via hydrogen bonds, resulting in a tightly-bound hydration layer [92–94]. This hydration layer serves as a physical and energetic barrier to prevent adsorption of not only protein, but cells and other biomolecules. This hydration layer is also helpful in maintaining the bioactivity of certain molecules [95]. Among various inorganic and organic hydrophilic polymers employed in biomedical devices, polyethylene glycol (PEG) and hydrogels are two that seem most promising [95,96].

### 3.1.1. PEG

Due to the commercial availability of PEG, researchers have a wide range of options in terms of the surface packing density and the chain length of PEG [97]. PEG can easily be immobilized onto a surface by in situ growth via pre-adsorbed initiation groups on the device surface [98]. The end-group of PEG can also be modified with a bioactive molecule of choice to achieve specific functionalities [95].

For example, Sun et al. designed their glucose biosensor by immobilizing glucose oxidase onto carboxymethyl-PEG-carboxymethyl film on the surface of glassy carbon electrode [81]. When tested against whole blood adhesion, they found that fibrins and blood cell deposits were drastically reduced. They measured blood glucose level using an ex vivo differential pulsed voltammetry (DPV, Table 1) and showed that their PEG-modified biosensors had good performance (LOD = 12.4 µM) and selectivity against ascorbic acid (AA) and uric acid (UA) (Table 2).

Hui et al. grafted PEG onto conductive polyaniline (PANI) nanofibers as a anti-biofouling substrate for breast cancer susceptibility gene biosensor. Their DPV-based biosensor showed a high sensitivity to targeted DNA with linear range from 0.01 pM to 1 nM. Their PEG-grafted polymer nanofibers had excellent anti-biofouling against protein solution and human serum. After incubation in 1%, 20%, or whole human serum, the response currents retained 99.05%, 93.93%, and 92.17% of their initial values [99]. However, the polymer chain formation of PEG can become unstable when temperature rises >35°C, which might affect its anti-biofouling capacity in vivo [100].

### 3.1.2. Hydrogels

Hydrogels are three-dimensional polymeric networks formed by chemical or physical cross-linking, which absorb and retain a large amount of water. This unique permeable structure enables easy access to analytes, and similar mechanical properties to soft body tissue. When incorporated with conductive polymers such as polypyrrole, poly(ethylenedioxythiophene) (PEDOT), and PANI, they can further facilitate electron transport for a higher biosensor sensitivity. The water content and porosity of hydrogels can be tuned by controlling the degree of cross-linking [98].

For example, Quinn et al. made glucose-permeable hydrogel (97% water by mass) by cross-linking amine-terminated PEG and the di-succinimidyl derivative of PEG propionic acid in water, at room temperature, and without an initiator [101]. After subcutaneous implantation in rats for 7 days,

their hydrogel had low cell deposits, while the Pellethane™control sample was encapsulated with tissue consisting of macrophages, neutrophils, foreign body giant cells, fibroblasts, and collagen. However, when using this PEG hydrogel as amperometric glucose biosensor interface for subcutaneous measurement in rats, there was a 34% loss in sensitivity.

Yu et al. made their hydrogels (146–217% water by mass) based on a copolymer of hydroxyethyl methacrylate (HEMA) and 2,3-dihydroxypropyl methacrylate (DHPMA) [82]. When used an anti-biofouling coating for a glucose biosensor made of platinum wire and glucose oxidase enzyme, hydrogels helped sensors subcutaneously implanted in rats maintain their functionality for 21–28 days. Histology also revealed that the fibrous capsulation surrounding hydrogel-coated sensors were 50–80% thinner than the ones coated with epoxy-polyurethane (Table 2). However, there are some concerns of poor mechanical properties, adhesion issues, and toxicity of chemicals used for cross-linking of hydrogels [102].

### 3.2. Zwitterionic Polymers

Zwitterionic polymers have equal number of anionic and cationic groups in their monomer chains, and they are highly hydrophilic and anti-biofouling [103]. Similar to hydrophilic materials, zwitterionic polymers prevent non-specific bonding via the formation of a hydration layer to serve as a barrier between biomolecules and their surfaces. Instead of hydrogen bonds, zwitterionic polymers form a hydration layer via ionic solvation [104–106]. Based on anion types (typical cations are quaternized ammonium), zwitterionic polymers can be categorized into sulfobetaine (SB), carboxybetaine (CB), or phosphorylcholine (PC) types. SB types have the best commercial availability due to their preparation simplicity. CB types have superb anti-biofouling property, biocompatibility, and functionality. PC types are excellently biocompatible, but have a high cost of production [107,108].

While zwitterionic polymers demonstrate good anti-biofouling properties, their poor conductivity often requires additional steps for them to be used in implantable biosensors [109]. To address this, Wu et al. combined SB with conductive polymer PEDOT for their glucose biosensors. They also encapsulated glucose oxidase within their zwitterionic polymer in a one-step electropolymerization method. Using quartz crystal microbalance (QCM), they demonstrated that their biosensor surfaces had excellent resistance to fibrinogen and human blood plasma adsorption, compared to the PEDOT counterpart. The sensitivity and the linearity did not show significant decline after incubation in plasma over 14 days (Table 2) [83].

Xu et al. combined CB with thiophene (Th) to form a linear zwitterionic conductive polymer PCBTh [110]. Further, by introducing 9'-bifluoreneylidene (9,9'-BF) distorted units into the polymer backbone, they synthesized another porous zwitterionic conductive polymer PCBTh-*co*-BF, which has additional nanoscale porous structure. Both zwitterionic polymers showed excellent anti-biofouling properties, evidenced by reduction of BSA attachment by 83.6% and 93.6% (Figure 2a), astrocyte cells attachment by 97.7% and 100.0%, and *Staphylococcus aureus* attachment by 56.9% and 74.0% (Figure 3a).

Similarly, Lee et al. developed their zwitterionic polymer (sulfobetaine methacrylate) (pSBMA) hydrogel, which showed 80% reductions of fibrinogen absorption and cell (human dermal fibroblasts, hDFBs) adhesion [111]. Chou et al. developed their zwitterionic hydrogel (Poly(carboxybetaine acrylamide) (pCBAA)), which can be coated using spin coating [112]. The thickness of the coating can be controlled between 15 and 150 nm by adjusting the crosslinker concentration. The coating demonstrated antifouling properties against undiluted human blood serum, and high antibody functionalization for specific biomarker detection (human thyroid stimulating hormone). When the coating thickness of their zwitterionic hydrogel was over 25 nm, protein adsorption from the undiluted human blood serum dropped to an ultra-low fouling level ($<10$ ng/cm$^2$).

Hu et al. coated an amperometric glucose biosensor with their zwitterionic pSBMA [113]. This anti-biofouling coating showed long-term stability in 37 °C undiluted bovine serum, diminishing over 99% non-specific protein adsorption and maintaining 94% of sensor sensitivity after 15 days.

The zwitterionic polymer coated biosensor also had nearly 50% reduction in sensitivity deviation compared to a polyurethane coated control. An in vivo study done by Zhang et al. demonstrated zwitterionic hydrogels resisted the formation of encapsulation for at least three months in mice subcutis [77]. They also reported zwitterionic hydrogels promote angiogenesis and macrophages phenotypes associated with anti-inflammatory, pro-healing functionality.

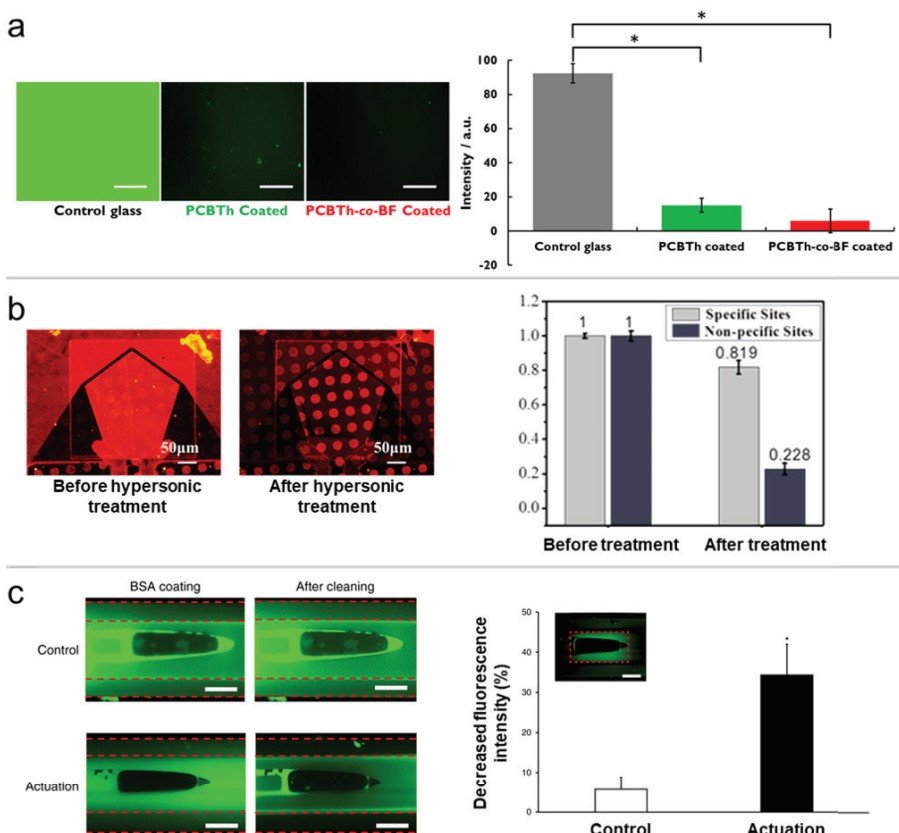

**Figure 2.** (**a**) Zwitterionic linear and porous polymers prevented protein binding to the surfaces. Scale bar = 200 μm. Adopted from [110]. (**b**) Using a hypersonic resonator, non-specific bound proteins were selectively removed. Adopted from [90]. (**c**) Proteins adsorbed to the actuators and surrounding tubing surfaces was removed during magnetic actuation. Scale bar = 200 μm. Adopted from [91].

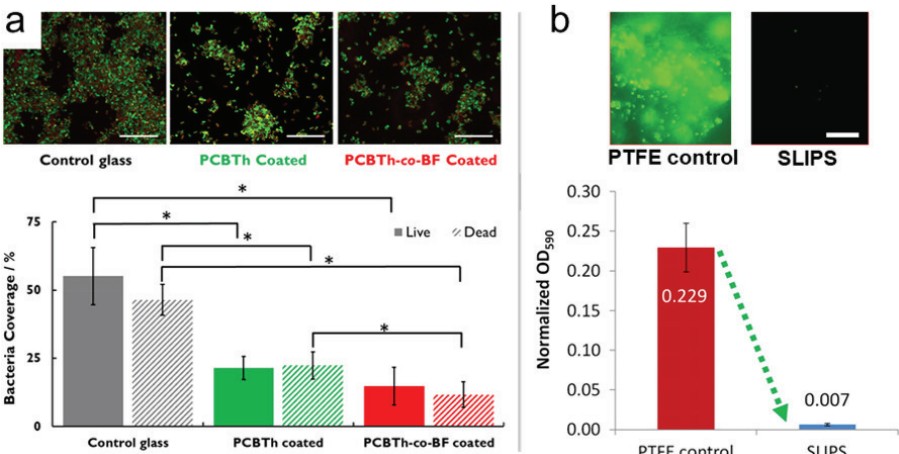

**Figure 3.** (**a**) Zwitterionic linear and porous polymers prevented bacteria growth on the surfaces. Scale bar = 20 μm. Adopted from [110]. (**b**) Superhydrophobic surfaces had excellent resistance to biofilm growth. Scale bar = 30 μm. Adopted from [114].

### 3.3. Naturally-Occurring or Bio-Mimicking Materials

Being similar to macromolecules that are readily available in the host body, naturally-occurring or bio-mimicking materials have the innate biocompatibility advantages because they can be recognized by the immune system and processed metabolically [102]. When used in biosensors, these materials can serve as sensor structure components or additional coating, usually immediately beneath the biorecognition component. Naturally-occurring materials include chitosan [115–117], collagen (Table 2) [84], hyaluronic acid [118], silk fibroin [115], gelatin [119], and lubricin [120], all of which showed good anti-biofouling performance in medical devices.

For application in biofouling prevention for biosensors, Burugapalli et al. developed electrospun membranes based on polyurethane and gelatin [119]. Although both membranes featured similar micron-sized pores, the membranes with gelatin prevented the fibroblasts and formation of fibrous capsulation while the polyurethane membrane failed to do the same. This anti-biofouling membrane helped prolong the in vivo glucose biosensing in rat subcutaneous tissue for at least three weeks.

In work done by Shrestha et al., chitosan-glucose oxidase was immobilized on polypyrrole-Nafion-functionalized multi-walled carbon nanotube nanocomposite. Their biosensors with naturally-occurring materials had improved sensitivity of 2860.3 $\mu A \cdot mM^{-1} \cdot cm^{-2}$, the LOD of 5 $\mu M$, and the selectivity against interference of AA, UA, and dopamine (DA) [116]. Naturally-occurring biomolecules such as amino acids, peptides, and polysaccharides can also be used as building blocks for novel bio-mimicking anti-biofouling materials. For example, some peptides have been used as anti-biofouling substrates for biosensors, demonstrating their capabilities to prevent non-specific protein adsorption in human serum (Table 2) [85,121]. However, naturally-occurring materials can suffer from several disadvantages including the difficulty of processing, high variability, and limited metabolic lifetime [102].

Interestingly, synthetic peptides, peptoids, are also being actively investigated for anti-biofouling functionalities. For example, Statz et al. designed their peptidomimetic polymer consisting of biomimetic anchoring peptides and antifouling peptoids, which resembled repeating units of PEG [122]. This biomimetic polymer resisted 3T3 fibroblast cell attachment and serum protein adsorption for over five months. Compared to unmodified surfaces, adsorbed protein layer on peptoid-modified titanium surfaces after 20 min exposure to whole human serum decreased from 435 to 5 ng/cm$^2$. Later, they designed more peptidomimetic polymers with different side-chains, showing methoxyethyl side-chain has superior long-term fouling resistance compared to hydroxyethyl and hydroxypropyl ones [123]. They also demonstrated the resistance against bacteria, in addition to protein (~99% decrease in fibrinogen adsorption) and cells (~98% decrease in 3T3 fibroblast cell density). Although they offer excellent design flexibility, peptides and peptoids based materials require a tedious and expensive synthesis process [96].

Other than the materials themselves, naturally-occurring surface morphologies also show great promise in anti-biofouling. Nanopillar structures found on wings of butterflies [124], dragonflies [125], cicadas [126], and skin of gecko [127], all showed excellent bactericidal property. Through experimental and modeling studies, researchers have suggested that the bactericidal property is achieved via physical-mechanical interaction between bacteria membrane and nanostructure, regardless of the type of the materials [126,128]. Once bacteria make contact with surface with nanostructures, tips of the nanopillars can act as anchor points for cell adhesion. Afterward, gravity and van der Waals forces induce excessive stretching of the bacteria membrane, leading to membrane rupture and cell lysis. Further studies summarized that nanopillars with approximately 50–250 nm in diameter, 80–250 nm in height, and 100–250 nm in spacing can provide these killing-upon-contact properties [129]. Inspired by these naturally-occurring surface morphologies, various surfaces such as silicon [130], metals [131,132], polymers [133] have been fabricated using various nanofabrication techniques.

While these biomimetic nanostructured surfaces showed great promise in bactericidal functionality, which leads to the decreased usage of antibiotics and bacterial drug-resistance, there are still some disadvantages for them to be used in implantable biosensors. First, the bacteria killed

on the surfaces can remain on the device surface and may still affect the functionality of the implantable biosensor. Therefore, other active approaches may be necessary to clear them away from biosensor surfaces. Second, the fabrication method of nanostructure on surfaces is often complicated, time-consuming, and limited to planar surfaces. This can result in increased cost and limited application. Lastly, there are not many in vivo studies to confirm whether these bactericidal surfaces can truly reduce the foreign body response [134].

Meanwhile, there are other non-biomimetic nanostructured surfaces that show improved biocompatibility with better immune and inflammatory responses [135–137]. In a recent study, Xu et al. used oxygen plasma to nanotexture both planar (i.e., films) and nonplanar (i.e., tubes) polytetrafluoroethylene (PTFE) surfaces for bactericidal and anti-inflammatory properties. Oxygen plasma has been used as a facile method to fabricate nanostructures on polymers before [138], but it is still limited to the planar surface. Here, by positioning tubes vertically in the middle region of the etching chamber, they achieved the fabrication of nonplanar large-scale nanostructures with excellent axial and radial uniformity for the first time. When tested in vitro, 30-min-etched tubes had 66.90% less live and 83.34% more dead *Staphylococcus aureus* coverage, compared to non-etched control tubes, showing bactericidal property. When implanted in mice subcutis, etched films also demonstrated anti-inflammatory and pro-healing properties, as evidenced by the thinner inflammatory band (Figure 4a), lowered collagen deposition, and decreased macrophage infiltration [139].

### 3.4. Superhydrophobic Surfaces

For a smooth solid surface, its wettability is determined by its chemical composition and surface energy. The lower surface energy the surface has, the more hydrophobic the surface becomes. The surface is considered to be hydrophilic or hydrophobic when the water contact angle $\theta$ is smaller or higher than 90°. Being the world's lowest surface energy material, PTFE theoretically has a water contact angle of 115.2° [140]. Moreover, when structures, especially nanostructures, are introduced to the surface, the surface could become superhydrophobic ($\theta$ higher than 145°). In these cases, the droplets partially contact with the surface, leaving air trapped between surface protrusions, transiting the state of wettability into Cassie's model [141]:

$$cos\theta = fcos\theta + (1 - f)cos180° = fcos\theta + f - 1, \qquad (4)$$

where $f$ accounts for the fraction of the whole solid surface to the corresponding smooth surface. In Cassie's model, the droplets on the surfaces can be easily removed (i.e., self-cleaning). Taking advantage of this, biomacromolecules adsorbed on the surface can be removed by a low shear force to achieve anti-biofouling properties.

For example, cross-linked hyperbranched fluoropolymers and PEG networks developed by Gudipati et al. had surface energy as low as 22 mJ/m$^2$ that were capable of resisting protein, lipopolysaccharide, and *Ulva* zoospore adhesion, as well as releasing (self-cleaning) zoospore and sporeling [142]. First developed by Epstein et al., slippery liquid-infused porous surfaces (SLIPS) showed excellent resistance to bacteria attachment and biofilm growth [114]. These surfaces were made by adding perfluorinated lubricating fluids onto porous PTFE films. SLIPS prevented nearly 100% of bacteria attachment (Figure 3b), making them far superior to PTFE and a range of nanostructured superhydrophobic surfaces.

Later, Leslie et al. developed a coating technique based on SLIPS, which involves the covalent binding of a flexible molecular tethered perfluorocarbon layer, followed by the coating of a layer of liquid perfluorodecalin [143]. After the coating, the surfaces became extremely omniphobic (repel both water and oil) and had anti-biofouling property against fibrin and platelet, as well as biofilm adhesion. In their in vitro study, coated acrylic and polysulfone surfaces reduced platelet adhesion by 27- and 4-fold, respectively, compared to the uncoated control surfaces. Additionally, coated polyvinyl chloride medical tubing showed an 8-fold reduction in *Pseudomonas aeruginosa* biofilm formation compared to

control tubing. In their in vivo study, arteriovenous implanted coated tubing in pigs remained potent for at least 8 h without anticoagulation. This could potentially offer a new way to prevent thrombosis in extracorporeal circuits without the complications of heparin anticoagulant therapy. However, these lubricant-infused omniphobic surfaces are challenging to be applied to biosensing, because they also prevent the adhesion of any biological recognition component as well as the target analyte.

To overcome this, Badv et al. developed their surfaces with self-assembled monolayers (SAMs) of aminosilanes and fluorosilanes [86]. Aminosilanes were used as coupling molecules for immobilizing antibodies, and fluorosilanes were used for lubricant infusion, enabling anti-biofouling against proteins, cells, and bacteria. Their fluorescent biosensors successfully detected the presence of endothelial cells not only in buffer but also in human plasma and human whole blood (Table 2).

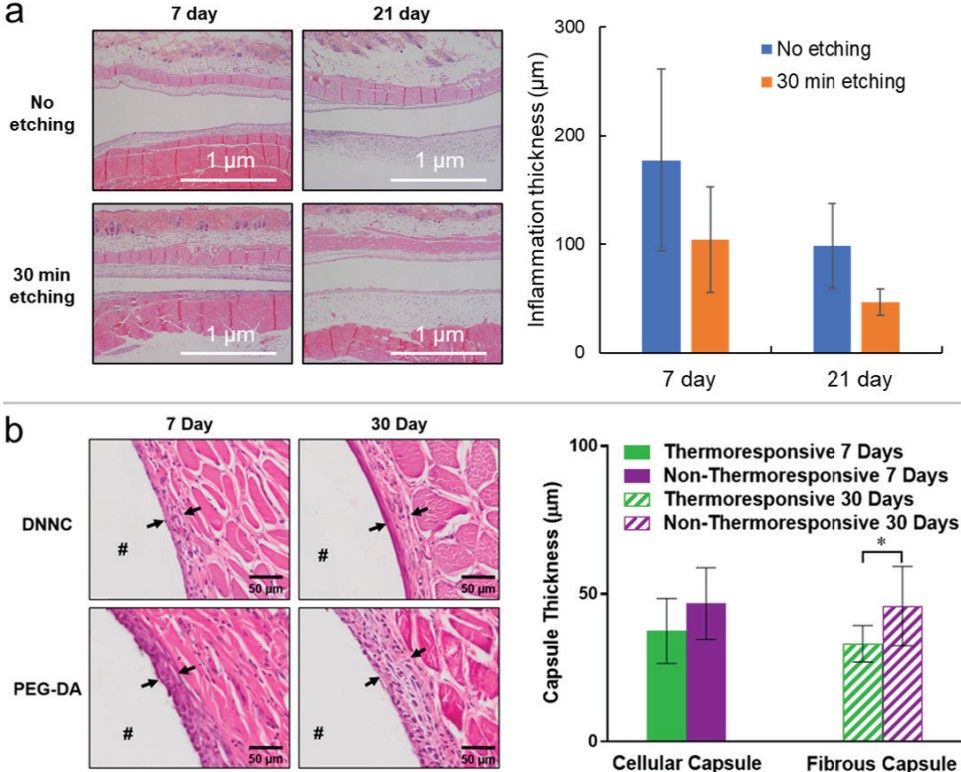

**Figure 4.** (**a**) Nanotextured polytetrafluoroethylene (PTFE) films after etching showed a thinner inflammatory band after 7 or 21 d of subcutaneous implantation. Adopted from [139]. (**b**) Implants made of temperature-responsive materials showed decreased fibrous capsulation thickness after 7 or 30 d of subcutaneous implantation. Adopted from [89].

## 3.5. Drug-Eluting Materials

The rapid technological developments of drug delivery enable researchers to develop polymers that are incorporated with therapeutics that can be passively released over time (i.e., drug-eluting). These drug-eluting materials have the potentials to regulate host response so that implantable devices can be better integrated with the body and have a longer lifetime. For example, sirolimus-eluting stents are widely known for minimizing restenosis, a process during which fibrosis can reduce and block the stented artery [144]. Using the same principle, implantable biosensors equipped with drug-eluting materials can regulate local foreign body response, have extended lifetime, and maintain high sensitivity, selectivity, and accuracy. Here, we will focus on three specific types of drugs that are actively explored in implantable biosensors research.

### 3.5.1. Nitric Oxide

Nitric oxide (NO) plays an important role in vasodilation, blood pressure, immune response, neural communication, and biofilm regulation [145]. It is also a potent anti-platelet agent. As a result, NO-eluting coatings including polyvinyl chloride, silicone rubber, polymethacrylate, and polyurethane could have potential in mitigating thrombosis and foreign body response in biomedical devices [146]. The NO-eluting can be achieved by using NO donors (e.g., diazeniumdiolates and nitrosothiols) or NO precursors (e.g., nitrite and nitrosothiols) in the presence of physiological reducing agents (e.g., ascorbate and thiolates).

For example, Schoenfisch et al. developed their gas permeable coatings using cross-linked silicone rubber containing diazeniumdiolates for amperometric oxygen sensors [147]. When implanted within the arteries of the dogs without administrating anticoagulation drugs, the NO-eluting biosensors had accurate measurement of partial pressure of oxygen for 14 h. In comparison, control sensors without NO-eluting had nearly 50% discrepancy with thenactual oxygen level due to thrombus formation on the sensor surface. Their collaborators also reported similar results when tested in carotid and femoral arteries of swine over 16 hours with NO eluting rate of $>1 \times 10^{-10}$ mol/cm$^2$·min [148]. They reported that the NO-eluting material had significantly lowered deposition of blood clots and platelets.

Hetrick et al. developed their sol-gel derived coatings (e.g., xerogels) capable of storing and spontaneously releasing NO [149]. When implanted subcutaneously, they observed that implants with NO-eluting coating had 50% decrease in collagen capsule after three weeks and 30% decrease in inflammatory response after six weeks, compared to the uncoated silicone elastomer. In their in vivo study, glucose biosensor with NO-eluting coatings showed better accuracy and faster response time due to reduced thrombus formation [150,151]. In rabbit veins over 8 h, their glucose biosensors with NO-eluting coating reported 97.5% accurate readings (evaluated using Clarke error grid), compared to 86.7% for control biosensors [150].

In another example, McCabe et al. achieved NO-eluding by embedding *S*-nitroso-*N*-acetylpenacillamine in the walls of catheter-type oxygen sensors [152]. Compared to the control, their novel sensors showed a better accuracy when continuously tested in rabbit and swine models for 20 hours and significantly less thrombus formation. In their latest study, the anti-inflammatory effect of NO can be further improved by combing transmembrane protein such as CD47 peptides [153]. Nevertheless, a chronic study is still needed to clarify the mechanism of NO suppressing foreign body response.

### 3.5.2. Anti-Inflammatory Drug

It is common to use steroidal and non-steroidal anti-inflammatory drugs after surgeries as a part of routine care for patients. Functionally, these drugs (e.g., glucocorticoid and dexamethasone) suppress the immune response by inhibiting inflammatory mediators such as vasoactive and chemoattractive factors, prostaglandins and leukotrienes, and lipolytic enzymes [72,102]. By inhibiting these mediators, few inflammatory cells become activated at the implant site, suppressing the fibroblast proliferation. Since long-term systemic use of these drug leads to unwanted side effects, local and continuous eluting from the implant itself may be useful for regulating the foreign body response to implanted biosensors and preventing fibrosis formation.

For example, Go et al. developed biodegradable poly (lactic-co-glycolic) acid (PLGA) microspheres capable of eluting anti-inflammatory α-melanocyte stimulating hormone (α-MSH) [154]. The α-MSH was released over 72 hours and reduced the expression of inflammatory cytokine and tumor necrosis factor-α (TNF-α). Their in vivo experiment showed that α-MSH coated PLGA reduced the activation of neutrophils and macrophages when subcutaneously implanted in rats.

Jayant et al. developed nanoengineered alginate microspheres for localized dexamethasone and diclofenac sodium release for an implantable fluorescent glucose biosensor [155]. Their microspheres were able to elute a drug for 30 days at a controlled rate. The in vitro and the in vivo study revealed that their materials were not cytotoxic and successfully suppressed inflammation at the implant site.

Later, when tested in simulated interstitial body fluid, their glucose biosensors showed good enzyme retention and bio-activity for one month (Table 2) [87].

Li et al. made their dexamethasone-loaded copolymer with good biocompatibility using hot melt extrusion method, which is wildly used in implantable device manufacturing [156]. PLGA/poly(vinyl alcohol) (PVA) composite hydrogel developed by Wang et al. successfully eluted dexamethasone and was used to coat a dummy glucose biosensor using the mold fabrication process [157]. In their animal study of one month, tissue implanted with drug-eluting materials remained normal (i.e., similar to healthy tissues), with no inflammatory reaction or fibrous encapsulation occurring.

### 3.5.3. Angiogenic Drugs

One failure mode for implantable biosensors is the total blockage of analyte to sensor surface due to the hypovascular encapsulation. To mitigate this, researchers have investigated strategies that promote angiogenesis near the implant site. Newly formed blood vessels can improve the analyte transport around the implanted biosensor and maintain its performance. The angiogenesis process can be achieved using drugs such as the vascular endothelial growth factor (VEGF). Another benefit of angiogenesis is that well-vascularized tissue at the implant site is also critical for healing the trauma caused by the implantation process [158].

For example, Ennett et al. studied the release of VEGF from poly(lactide-co-glycol-ide) (PLG) scaffolds or microspheres [158]. Their in vivo studies revealed that eluted VEGF had local concentrations above 10 ng/mL at distances up to 2 cm from the implant site for 21 days without entering systemic circulation, and significantly enhanced the local angiogenesis. Ward et al. measured the density of microvessels close to a model disk biosensor implanted subcutaneously after 40 days with 28-day continuous infusion of VEFG or saline control. VEGF treated tissues 1 mm away from the implant site had nearly 300% higher capillary density compared to the control [159]. This neovascularization happened 13 mm away but not 25 mm away, demonstrating the desired local effectiveness. Further, they examined the performance of amperometric glucose sensor implanted in rat with or without local subcutaneous infusion of VEGF via osmotic pumps. Results show that the biosensor implanted 2 mm away from the VEGF infusion had a faster response time and a higher accuracy compared to the biosensor implanted 15 or 22 mm away (Table 2) [88].

VEGF-eluting materials have also been used concurrently with anti-inflammatory drugs. When corticosteroid drugs are used to lower the inflammation, they also downregulate endogenous VEGF that inhibits angiogenesis [160], making this a two-pronged approach (i.e., control of inflammation and induction of angiogenesis at the same time) necessary [161]. For example, Sung et al. studied the average vascular density of the chick embryo chorioallantoic membrane tissue implanted with a hydrogel-coated microdialysis capillary sensor with simultaneous ($1.24 \pm 0.35 \times 10^{-3}$ vessels/mm$^2$), sequential ($1.15 \pm 0.30 \times 10^{-3}$ vessels/mm$^2$), or no delivery ($0.71 \pm 0.20 \times 10^{-3}$ vessels/mm$^2$) of dexamethasone and VEGF for eight days. Calculation of vasculature/inflammation ratio for sequential dexamethasone/VEGF delivery was 60.3% and 139.3% higher than that of VEGF and dexamethasone release alone, respectively, and was also 32.1% higher when compared to simultaneous administration [162].

In addition to NO, anti-inflammatory, and angiogenic drugs, surfaces eluding heparin have also shown excellent anti-biofouling properties. Being anticoagulant, heparin has long been used in medical devices to improve their blood compatibility [163]. Research groups have previously developed PVA or polyethylene terephthalate (PET) based hydrogels that can continuously elude heparin for three weeks [164,165]. Such heparin-eluding or heparin-mimicking materials have been demonstrated in biosensors for anti-biofouling [166,167]. For example, Sun et al. prepared their glucose amperometric biosensors using polyurethane-heparin nanoparticles. Their biosensors had LOD of 14 μM when tested in whole blood, and good selectivity against AA and UA [168].

## 4. Active Anti-Biofouling Strategies

Smart or stimuli-responsive materials are the types of materials that can reversibly change their properties in a controlled fashion via external stimuli such as stress, light, pH, temperature, electrical or magnetic field [169]. These materials have been extensively used and researched for wide-ranging applications including drug delivery, sensors, actuators, microfluidics, and oil/water separation [170–173]. In this section, we will explore various anti-biofouling materials that respond to temperature or pH. Moreover, we will discuss other stimuli-response materials and transducers that have been developed for anti-biofouling applications.

### 4.1. Temperature-Responsive

Due to the circadian rhythm, the human body temperature can vary hour-by-hour and day-by-day. To take advantage of these normal body temperature fluctuations, researchers have developed smart materials based on poly(*N*-isopropylacrylamide) (PNIPAAm) that can swell and deswell when they are above and below volume phase transition temperature (VPTT). This periodic change in material volume may be used to remove adhered cells and proteins.

For example, Gant et al. prepared their thermoresponsive nanocomposite hydrogel using photopolymerization of an aqueous solution of *N*-isopropylacrylamide and polysiloxane colloidal nanoparticles. They also added *N*-vinylpyrrolidone as a co-monomer to increase VPTT and improve mechanical strength so the hydrogel can be more suitable as a glucose biosensor coating. In their in vitro study over seven days, their hydrogels that were exposed to thermal cycling exhibited a very low level of GFP-H2B 3T3 mouse fibroblast cells attachment compared to the polystyrene control [174].

Similarly, Fei et al. made their thermoresponsive double network hydrogel membranes by combining PNIPAAm with electrostatic co-monomer, 2-acrylamido-2-methylpropane sulfonic acid [175]. They evaluated the glucose diffusion kinetics, thermosensitivity, and cytocompatibility of their hydrogel, and demonstrated the self-cleaning functionality towards mouse mesenchymal progenitor 10T1/2 cells. Later, in two animal studies [89,176], thermoresponsive PNIPAAm hydrogels implanted subcutaneously in rodents showed faster healing, thinner fibrous capsulation (Figure 4b), and increased microvascular density, compared to benchmark PEG control (Table 2) [89].

### 4.2. pH-Responsive

Materials that are responsive to environment pH values may also be used in drug-eluting coatings for anti-biofouling and suppressing inflammation. During foreign body response, macrophages release phagolysosome in the process of phagocytosis, which has acidity as low as pH 4, making the tissues near implants to have a relatively low pH value [177]. By utilizing this phenomenon, anti-inflammatory drugs may be eluted to suppress the inflammation and potentially extend the lifetime of implantable biosensors.

For example, Wang et al. developed a pH-sensitive molecularly imprinted polymer nanospheres/hydrogel composite as a coating for implantable glucose sensors [178]. Their coating exhibited a pH-responsive release profile for dexamethasone-21 phosphate disodium and a controlled release over a six-week period. Their hydrogel composite exhibited a faster release rate at a lower pH value within the pH range tested (6.0–7.4), which is desirable for suppressing inflammation. In work done by Ninan et al., novel carboxylated agarose/tannic acid hydrogel scaffolds cross-linked with zinc ions were designed for the pH-controlled release of tannic acid [179]. The elution of tannic acid only happened at acidic pH, where hydrogel displayed swelling. Their in vitro evaluation also revealed anti-bacterial property, a lack of cytotoxicity towards 3T3 fibroblast cell, significantly greater cell migration and proliferation, and suppression of NO production, indicating effective anti-inflammatory functionality.

### 4.3. Surfactant-Desorbing Surfaces

Reported by Xu et al., dodecylbenzenesulfonate-doped polypyrrole (PPy(DBS)) surfaces have been shown to switch its wettability in situ, allowing attached oil droplets to roll away with minimal tilting [180,181]. This self-cleaning functionality is realized by the elution of surfactant dopant during the electrochemical reduction. Surfactants decrease the surface tensions at both oil/water and water/solid interfaces, dramatically reduce the work of adhesion between the surface and the fouling agents, enabling self-cleaning.

While there are not many studies regarding the use of surfactant-eluting surfaces for anti-biofouling, previous studies have suggested that proteins adsorbed on the surfaces may be effectively removed by adding surfactants, especially the anionic ones [182]. For example, using in situ ellipsometry, Wahlgren et al. observed the removal of adsorbed $\beta$-Lactoglobulin from three hydrophobic materials (silica, chromium oxide, and nickel oxide) and one hydrophilic material (methylated silica). After adding 3.68 g/mL anionic surfactant sodium dodecyl sulfate (SDS), plateau amount of adsorbed protein decreased to 0–0.12 $\mu$g/cm$^2$ [183].

Moreover, Kumar et al. deposited a cholesterol oxidase enzyme and a potassium ferricyanide mediator on PPy(DBS) surfaces using a physical adsorption technique to form an amperometric cholesterol biosensor [184]. Their sensor showed linearity over the range of 2–8 mM cholesterol solution, response time of 30 s, and stability for three months at 4 °C. Besides using electropolymerization, PPy(DBS) can be made via chemical oxidization, allowing its inkjet-printing. Such direct writing technique is key to make flexible biosensors that are capable of measuring multiple analytes at the same time [7,8,185,186].

### 4.4. Acoustic Waves

Piezoelectric materials generate electrical charge when pressure was applied to them. When applied with electrical potential of certain high frequency, piezoelectric materials can vibrate and generate acoustic waves, which in turn can induce shear stress on the surfaces. Using this phenomenon, biosensors consisting of piezoelectric materials have been proposed to use acoustic waves to clear the non-specific protein adsorption. For example, Yeh et al. studied the protein removal from a lead zirconate titanate (PZT) composite with natural frequency of 16 kHz. In their tests, PZT plates were first incubated with BSA or anti-mouse IgG and underwent different vibration conditions (i.e., voltage, frequency, duration). Their results suggested a nearly 50% protein removal due to the acoustic streaming [187].

In finite element simulations done by Sankaranarayanan et al., surface acoustic-waves propagating on a lithium niobite piezoelectric crystal were studied to evaluate the effects of applied voltage intensity, device frequency, fluid viscosity, and density on the removal of non-specific bound proteins [188]. Their results suggested that the most effective protein removal happened when the generated acoustic wavelengths were close to the protein radius. In the work by Pan et al., non-specific bound proteins were removed by micro-vortexes induced by the microfabricated hypersonic resonator with 2.5 GHz resonant frequency [90]. They also showed the non-specific protein removal did not affect the specific antibody-antigen binding (Figure 2b), meaning such approach will not compromise the performance of the potential biosensors (Table 2).

### 4.5. Magnetic Actuation

In the work done by Lee et al., magnetic microactuators were incorporated into chronically implantable catheters to remove the biofouling. For hydrocephalus patients, a shunt system is required to drain excess cerebrospinal fluid to relieve the increased intracranial pressure. Unfortunately, these shunt systems failed frequently due to the obstruction of the ventricular-catheter pores induced by the accumulation of cellular materials. Using magnetic microactuators that can fit inside the catheter pores, they showed that the adhered cells can be removed wirelessly, achieving on-demand,

in situ anti-biofouling [189]. Such microactuators are mechanically robust so they can be implantable long-term and activated regularly to remove the biofouling [190]. The microscale magnetic component has also been shown to be compatible with magnetic resonance imaging [191]. Further work done by Yang et al. showed these microactuators can effectively remove 90% of non-specific adsorbed protein, in addition to the removal of cells [192].

In another case, Park et al. developed an implantable glaucoma drainage device with self-clearing ability by combining similar magnetic microactuators [91]. Using finite element modeling, they showed that actuation-induced shear stress near the perimeter of the actuator can be as high as 10 dyn/cm$^2$, enough to clear protein biofouling. In vitro test showed 85% of BSA previously coated on the actuator was cleared, evidenced by the decrease of fluorescent intensity (Table 2 and Figure 2c) [91]. In addition, there have been other examples of developing anti-biofouling magnetic actuators using magnetic nanoparticle-infused surfaces [193–195].

Recently, there have been several attempts to develop multifunctional "smart catheters" [196]. In additional to serving its designed catheter function, an attached biosensor can in situ measure multiple analytes. For example, Li et al. designed a smart catheter for patients with traumatic brain injury that had integrated oxygen, glucose, and temperature biosensors [197]. Biosensors were fabricated on Kapton film, which were spirally rolled onto a proof-of-concept intraventricular catheter with inner diameter of 1.5 mm and outer diameter of 1.7 mm. All three sensors demonstrated stable measurements of respective analytes in cerebrospinal sample fluid with high accuracy. They also developed a program to compensate for the effects of temperature on oxygen biosensor performance, as well as the effects of temperature and oxygen on glucose biosensor. In work done by Nacht et al., insulin infusion catheter was combined with optical glucose and oxygen biosensor [198]. Oxygen and glucose sensitive dyes were coated on the catheter, which were read by a two-channel phase fluorometer measurement module. Results from in vivo measurements in pig subcutaneous tissue well agreed with reference values obtained from the direct blood measurement. Although none of these multifunctional smart devices have demonstrated long-term reliability, it is not difficult to imagine a more prolonged functionality with the incorporation of these anti-biofouling strategies.

## 5. Conclusions

Over the decades, different types of biosensors have been developed and used to measure various analytes including glucose, lactate, neurotransmitters, ions, heavy metals, etc. These important clinical and research tools have drastically improved patient care with more accurate diagnosis and personalized therapy. The implantable biosensors have enormous potential to provide scientists and clinicians with a high-fidelity clinical data on physiological changes before, during, and after disease diagnosis and treatment. However, the existing implantable biosensors have limited long-term reliability, which poses a critical engineering challenge to unlock the full potential of implantable biosensors.

One way to improve the reliability of implantable biosensors is to mitigate biofouling and the foreign body response. There is already a large body of work to describe various passive and active approaches to address biofouling in vivo. Passive anti-biofouling strategies include the usage of various novel materials including hydrophilic, biomimetic, drug-eluting, and zwitterionic polymers. Active anti-biofouling strategies include the usage of stimuli-response materials, electromechanical and electromagnetic transducers. However, many of these reports are stand-alone attempts to demonstrate feasibility in vitro without the system integration and chronic in vivo evaluations. Therefore, the field is ripe with opportunities to demonstrate the long-term in vivo viability of anti-biofouling biosensors using any of the aforementioned and other approaches.

For example, we can envision a more reliable biosensor using a combination of passive and active anti-biofouling strategies to (1) delay the biofouling process and (2) to remove the bioaccumulation on demand. There is also the possibility to use multiple combinations of different materials, morphologies, and transducers to achieve a better long-term reliability. There are certainly other application-specific

and manufacturing challenges associated with translating these novel approaches for clinical use. For a passive anti-biofouling approach, a thorough investigation of the material or surface stability, scalability, and manufacturability will be critical. In addition, the frequency and the duty cycle of biofouling removal will need to be investigated for an active anti-biofouling approach. There are certainly many other critical engineering challenges in an effort to improve the reliability of implantable biosensors including the stability of the biorecognition elements and other abiotic failure modes. Nevertheless, addressing the biofouling-related failure mode is a huge step towards more chronically functional biosensors.

**Funding:** This work was supported in part by the National Science Foundation (United States) under Grants ECCS-1944480.

**Acknowledgments:** The authors would like to thank Jiwon Choi, Myriam Horsz, and Ángel Enríquez for their assists with manuscript preparation.

**Conflicts of Interest:** The authors declare no conflict of interest.

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
