# Peer review of "Anti-Biofouling Strategies for Long-Term Continuous Use of Implantable Biosensors"

_chemosensors, doi:10.3390/chemosensors8030066_

Round 1
Reviewer 1 Report
This is a useful and timely review. The coverage of the sequence of biofouling and the literature review of materials used to modify sensor surfaces will be particularly useful to researchers.
There are a handful of corrections and clarifications that are required before publication:
1) Eqn 3, the MARD definition cannot be right. I thing a summation sign is needed.
2) Eqn 2 needs clarification in the text. Sigma is the standard deviation of the signal, or the standard deviation of the ordinate intercept in a calibration working curve (the IUPAC definition). It also needs to be clearly differentiated from the limit of quantification.
3) line 56: four figures of merit are referred to but four have not been listed by this point.
4) Some discussion of how selectivity can be quantified would be a useful addition here.
5) Dynamic range and means of estimating it needs to be added to the introductory matter.
7) p5, line 135 "third" is misspelt.
8) ll 154-5: There is missing verb. Should this be "...expected to be..."
Author Response
Thanks for thorough review of our work. Please see the following point-by-point response to the reviewer comments.

Reviewer 2 Report
The review paper of “Anti-biofouling Strategies for Long-term Continuous Use of Implantable Biosensors” by Xu et al. well summarized the developing progress of anti-biofouling strategies for the fabrication of the implantable biosensors. The authors first presented the state-of-the-art implantable biosensors, and their design, and working principles. Then the various passive and active approaches to ameliorate biofouling were stated. The review is well written and will be of interest to the readers of the related fields. It is suggested to be published in the Chemosensors. The minor suggestions are as follows:
- The definition of N in equation (3) should be given.
- Some scheme concerning the design concept of anti-biofouling strategies of the implantable biosensors are suggested to be presented in the Review to make the content being better understood.
- The perspective of the anti-biofouling research of the implantable biosensors should be discussed in the Conclusion part.
Author Response

(The authors gave the same response as above.)

Reviewer 3 Report
This is a very well written review on materials and strategies for biosensors which resist bio-fouling to enable longer-term use. My largest critique is the authors state in the 2nd sentence of the paragraph that many biosensors developed in labs struggle when used in vivo, however, much of the presented data is from exactly those scenarios. There is no discussion of what research directions are needed to enable better long-term in vivo performance, only a review of what has already been demonstrated in very small scale experiments.
This does not detract from the main contribution of this work as a review paper.
The paper is easily understood but some minor fixes to grammatical errors are suggested. E.g. P3.81, P4.118, P6.195, P7.240. There also seem to be many more minor errors introduced after page 13 which are not enumerated here.
Finally, a minor point related to 4.3 Acoustic Waves: piezoelectric materials do not generate electricity from pressure, they generate charge when stressed. Pressure induces stress but so does bending or other material deformations. Perhaps this first sentence can be clarified by replacing electricity with charge and pressure with stress.
Author Response

(The authors gave the same response as above.)
